# Synthesis of Poly(Malic Acid) Derivatives End-Functionalized with Peptides and Preparation of Biocompatible Nanoparticles to Target Hepatoma Cells

**DOI:** 10.3390/nano11040958

**Published:** 2021-04-09

**Authors:** Clarisse Brossard, Manuel Vlach, Elise Vène, Catherine Ribault, Vincent Dorcet, Nicolas Noiret, Pascal Loyer, Nicolas Lepareur, Sandrine Cammas-Marion

**Affiliations:** 1University Rennes, Ecole Nationale Supérieure de Chimie de Rennes, CNRS, ISCR, UMR 6226, ScanMAT, UMS2001, F-35000 Rennes, France; clarisse.brossard@ensc-rennes.fr (C.B.); vincent.dorcet@univ-rennes1.fr (V.D.); nicolas.noiret@ensc-rennes.fr (N.N.); 2INSERM, INRAE, Institut NUMECAN (Nutrition Metabolisms and Cancer) UMR_A 1341, UMR_S 1241, University Rennes, F-35000 Rennes, France; manuel.vlach@univ-rennes1.fr (M.V.); elise.vene@univ-rennes1.fr (E.V.); catherine.ribault@univ-rennes1.fr (C.R.); 3Pôle Pharmacie, Service Hospitalo-Universitaire de Pharmacie, CHU Rennes, F-35033 Rennes, France; 4Comprehensive Cancer Center Eugène Marquis, F-35000 Rennes, France

**Keywords:** poly(malic acid) derivatives, GB virus A, circumsporozoite protein of *Plasmodium berghei*, (Co)polymers end-functionalized with peptide, multifunctional nanoparticles, human hepatoma cells

## Abstract

Recently, short synthetic peptides have gained interest as targeting agents in the design of site-specific nanomedicines. In this context, our work aimed at developing new tools for the diagnosis and/or therapy of hepatocellular carcinoma (HCC) by grafting the hepatotropic George Baker (GB) virus A (GBVA10-9) and *Plasmodium circumsporozoite* protein (CPB)-derived peptides to the biocompatible poly(benzyl malate), PMLABe. We successfully synthesized PMLABe derivatives end-functionalized with peptides GBVA10-9, CPB, and their corresponding scrambled peptides through a thiol/maleimide reaction. The corresponding nanoparticles (NPs), varying by the nature of the peptide (GBVA10-9, CPB, and their scrambled peptides) and the absence or presence of poly(ethylene glycol) were also successfully formulated using nanoprecipitation technique. NPs were further characterized by dynamic light scattering (DLS), electrophoretic light scattering (ELS) and transmission electron microscopy (TEM), highlighting a diameter lower than 150 nm, a negative surface charge, and a more or less spherical shape. Moreover, a fluorescent probe (DiD Oil) has been encapsulated during the nanoprecipitation process. Finally, preliminary in vitro internalisation assays using HepaRG hepatoma cells demonstrated that CPB peptide-functionalized PMLABe NPs were efficiently internalized by endocytosis, and that such nanoobjects may be promising drug delivery systems for the theranostics of HCC.

## 1. Introduction

At the beginning of the 20th century, Paul Ehrlich coined the term “Magic Bullet”, postulating that the targeting of drug towards specific diseased cells/tissues could be achieved without affecting healthy ones [1]. Ever since, tremendous research work has been done for designing efficient site-specific drugs using various ligand molecules, such as monoclonal antibodies (mAbs) [2,3,4], vitamins [2,5,6,7], carbohydrates [2,8], and aptamers [9,10,11]. More recently, short synthetic peptides have gained interest as targeting agents in the design of site-specific nanomedicines for therapy and/or diagnosis of several pathologies, especially cancers [2,12,13,14,15,16].

The discovery of peptides dates back to the turn of the 20th century, when Emil Fisher first synthesized in 1901 what he called a dipeptide, the glycyl-glycine, followed a few years later (1907) by the synthesis of an octadecapeptide, the Leu-Gly_3_-Leu-Gly_3_-Leu-Gly_9_ [17]. Almost in parallel, in 1902 Theodor Curtius synthesized benzoyl-glycyl-glycine using the azide reaction he developed [17]. Following these pioneer works, researchers worked to increase the number of amino acids in the peptide’s chain, which was possible thanks to the use of the removable benzyloxycarbonyl protecting group developed by Max Bergmann and Leonidas Zervas [17]. The uses of peptides as therapeutic molecules emerged in the 1950s when some pure peptides (oxytocin, vasopressin, for example) were synthesized by Vincent du Vigneaud [18,19,20]. However, their synthesis was long and tedious until the development by Merrifield of a functionalized resin, bearing his name, composed by styrene/divinylbenzene block copolymers, thus allowing faster and easier solid-phase peptides’ synthesis [18,19]. Since then, peptides (and proteins) are used for several indications under various therapeutic classes as a result of their high activity and specificity, and their low systemic toxicity resulting from their short in vivo half-life [18,21,22,23]. However, their fast degradation by peptidases and poor biodistribution at the targeted tissues/organs limit their uses. To overcome such drawbacks, strategies have been developed to increase peptides’ stability in vivo and to improve their biodistribution, including introduction of non-natural amino acids (D-amino acids, for example), cyclization, N- and C-termini and side-chain chemical modifications, etc. [21,22,23,24].

Beside their uses as therapeutic molecules [18,22,23,25], short peptides have been quite recently developed as targeting agents, since it was shown that peptide receptors are overexpressed, or even exclusively expressed, in tumors [12]. Such increasing interest has been catalyzed by the discovery, at the end of the 1980s, of specific receptors to somatostatin (SSTR1-5) expressed by human neuroendocrine tumors (NETs) and their metastases [12,26]. Octreotide, a somatostatin analog, grafted to diethylenetriamine pentaacetic acid (DTPA) chelating indium 111 (^111^In) was approved by the US Food and Drug Administration (FDA) in 1994 under the name of OctreoScan^®^ for the diagnosis and localization of NETs overexpressing the somatostatin receptor SSTR2 [12,27]. Twenty years later (2018), Lutathera^®^, which consists of octreotate, another somatostatin analog, grafted to 1,4,7,10-tetraazacyclododecane-1,4,7,10-tetraacetic acid (DOTA) chelating the lutetium 177 (^177^Lu), was the first FDA-approved radiopharmaceutical for the therapy of SSTR2-positive NETs [12,28]. Since the emergence of somatostatin and its analogs, numerous peptides have been identified by different technologies, such as phage-display, for their high degree of specificity towards cancer cells [12,13,14,15,16]. Such peptides, which are usually easily synthesized on large scales at a low cost, can be chemically modified to be grafted on various materials, such as lipids or polymers, to engineer nanocarriers characterized by their good biocompatibility in addition to their excellent targeting properties [12,13,14,15,16,17].

The advanced stages of hepatocellular carcinoma (HCC), the major primary liver cancer, still have a very poor prognosis despite the recent progress in the therapeutic management of this cancer [29]. In addition, only a few patients are eligible for curative treatments in the early stages of the disease, or can be treated downstream to decrease the stage of their tumor so that they may be eligible for curative treatments [30,31]. Consequently, there is an urgent need for tools allowing effective therapies and/or early diagnosis of HCC.

In this context, the use of peptides able to specifically target hepatoma cells to detect as soon as possible even small nodules and/or treat early and advanced stages of HCC might be of great interest. In order to identify peptides exhibiting a strong tropism towards human hepatoma cells, in a previous work, we elected 12 peptides reported in the literature to bind hepatoma cells, including GB virus A (GBVA10-9) and *Plasmodium circumsporozoite* protein (CPB), and launched a comparative study of the in vitro internalization by human hepatocytes in primary culture and hepatoma cells of the corresponding biotinylated peptides binding fluorescent streptavidin to form a model bioconjugate [32]. The GBVA10-9- and CPB-streptavidin complexes showed the strongest binding to hepatoma cells [32]. We next demonstrated that the same biotinylated peptides, when bridged via the streptavidin to biotinylated nanoparticles (NPs) constituted by the amphiphilic block copolymer biotin-poly(ethylene glycol)-*block*-poly(benzyl malate) (BiotPEG-*b*-PMLABe), significantly enhanced the cell uptake by human hepatoma cells but not by normal human hepatocytes [32]. These data suggested that GBVA10-9 and CPB peptides were good candidates to functionalize polymeric nanoparticles to efficiently target human hepatic cancer cells.

Our main objective of the present study was to establish the synthesis of peptides end-functionalized poly(benzyl malate) and poly(ethylene glycol)-*block*-poly(benzyl malate) through a thiol/maleimide addition reaction in order to eliminate the streptavidin used in our previous study [32]. The corresponding nanoparticles varying by the nature of the peptide (GBVA10-9, CPB or their scrambled control counterparts, Figure 1) present at their surfaces and by the absence or presence of poly(ethylene glycol) were also prepared and characterized by dynamic light scattering (DLS) (Malvern PANalytical Ltd., Malvern UK), electrophoretic light scattering (ELS) (Malvern PANalytical Ltd., Malvern UK) and transmission electron microscopy (TEM) (JEOL, Tokyo, Japan). Moreover, a fluorescent probe (DiD Oil) was encapsulated and preliminary in vitro internalisation assays using HepaRG hepatoma cell lines were performed.

## 2. Materials and Methods

### 2.1. Materials and Apparatus

All chemicals were used as received. α-maleimide, ω-carboxylic acid PEG_62_ (Mw = 3000 g/mol, n = 62) and α-methoxy, ω-carboxylic acid PEG_42_ (Mw = 2015 g/mol, n = 42) were purchased from Iris Biotech GmbH (Marktredwitz, Germany). Tetraethylammonium hydroxide and 6-maleimidohexanoic acid were purchased from Sigma-Aldrich (Saint-Louis, MO, USA). Peptides were provided by Eurogentec (Liege, Belgium). 1,1′-dioctadecyl-3,3,3′,3′-tetramethylindo dicarbocyanine perchlorate (DiD Oil) was purchased from Invitrogen (Thermo Fisher Scientific, Illkrich-Graffenstaden, France). Dry THF was purchased from Acros Organics (Thermo Fischer Scientific, Illkrich-Graffenstaden, France). Dimethylformamide (DMF) was purchased from Sigma-Aldrich (Saint-Quentin-Fallavier, France). Spectra/Por 4 Dialysis Membrane was acquired from Repligen (Breda, The Netherlands). The monomer, benzyl malolactonate (MLABe), was synthesized as described previously [33].

***Nuclear Magnetic Resonance spectroscopy:*** The standard temperature was adjusted to 298 K. NMR spectra were recorded on a Bruker Avance III 400 spectrometer (Bruker, Wissembourg, France) operating at 400.13 MHz for ^1^H, equipped with a BBFO probe with a Z-gradient coil and a GREAT 1/10 gradient unit. The zg30 Bruker pulse program was used for 1D ^1^H NMR, with a TD of 64 k, a relaxation delay d1 = 2 s and 8 scans. The spectrum width was set to 18 ppm. Fourier transform of the acquired FID was performed without any apodization in most cases.

***Infrared spectroscopy**:*** FT-IR spectra were recorded on an Avatar 320FT-IR Thermo Nicolet spectrometer (Nicolet, Montigny le Bretonneux, France) between 500 and 4000 cm^−1^ by direct measurement.

***Size Exclusion Chromatography**:*** Weight average molar mass (Mw) and dispersity (Đ = Mw/Mn) values were measured by size exclusion chromatography (SEC) in THF at 40 °C (flow rate = 1.0 mL/min) on a GPC2502 Viscotek apparatus (Malvern Instruments Ltd., Malvern, UK), country) equipped with a refractive index detector Viscotek VE 3580 RI (Malvern Instruments Ltd., Malvern, UK), a guard column Viscotek TGuard (Malvern Instruments Ltd., Malvern, UK), Org 10 × 4.6 mm, a LT5000L gel column (for samples soluble in organic medium) 300 × 7.8 mm and a GPC/SEC OmniSEC Software (OmniSEC™ 4.6.1, Malvern Instruments Ltd., Malvern, UK). The polymer samples were dissolved in THF (2 mg/mL). All elution curves were calibrated with polystyrene standards.

***Dynamic Light Scattering and Electrophoretic Light Scattering****:* Dynamic light scattering (DLS) and zeta potential (electrophoretic light scattering, ELS) measurements are performed on a Nano-sizer ZS90 (Malvern Instruments Ltd., Malvern, UK) at 25 °C, with a He-Ne laser (Malvern Instruments Ltd., Malvern, UK) at 633 nm and a detection angle of 90 °C.

***Transmission electron microscopy****:* Transmission electron microscopy (TEM) images were recorded using a Jeol 2100 microscope (JEOL, Tokyo, Japan) equipped with a Glatan Orius 200D camera using a 80 KeV accelerating voltage on the THEMIS platform (ISCR—Rennes) (Univ Rennes, Rennes, France). Each sample was deposited on a Formvar-carbon film coated on a 300-mesh copper grid. After 6 min, the excess of the sample was removed and staining was realized with phosphotungstic acid (1 v%).

***Flow cytometry****:* Cells were analyzed by flow cytometry (Becton Dickinson LSRFortessa™ X-20), using the cytometry core facility of the Biology and Health Federative research structure Biosit, Rennes, France, to quantify the fluorescence emitted by the DiD Oil-loaded NPs within cells. Cytometry data were analyzed using FACSDiva^TM^ software (Software version 8.0.1, Beckton Dickinson, East Rutherford, NJ, USA).

***Confocal microscopy****:* Images of NPs in human HepaRG hepatoma cells were acquired with Leica TCS SP8 confocal microscope piloted by the Las AF software (Software version 3.3, Leica Microsystems, Wetzlar, Germany) (microscopy Rennes imaging center facility of the Biology and Health Federative research structure Biosit, Rennes, France) and processed with the ImageJ software (Open Source Software version 1.51n, Laboratory for Optical and Computational Instrumentation, University of Wisconsin, Madison, WI, USA).

### 2.2. Synthesis of PMLABe Derivatives

*Synthesis of tetraethylammonium 6-maleimidohexanoate (MalHexCOO*^− +^NEt_4_): The methanol of 947 μL (1.42 × 10^−3^ mol) of tetraethylammonium hydroxide solution (25 wt% in methanol) was evaporated under vacuum, and 4 mL of anhydrous chloroform was added. In parallel, 300 mg (1.42 × 10^−3^ mol) of 6-maleimidohexanoic acid were solubilized in anhydrous chloroform (4 mL) under N_2_ stream, and the flask was put in an ice bath. The chloroform solution of tetraethylammonium hydroxide was then added to this solution, the mixture was stirred for 1 h, and the chloroform was evaporated under vacuum leading to the expected tetraethylammonium maleimidohexanoate as a violet oil (Yield: 93%) that was characterized by ^1^H NMR. ^1^H NMR (400 MHz, Acetone-*d*_6_), δ (ppm): 1.26–1.34 (m, 2H); 1.35–1.36 (m, 12H); 1.53–1.61 (m, 4H); 2.25–2.29 (t, 2H); 3.44–3.48 (t, 2H); 3.51–3.56 (d, 8H); 6.84 (s, 2H).

*Synthesis of tetraethylammonium* α*-methoxy-*ω*-carboxy poly(ethylene glycol) (MeOPEG_42_-COO*^− +^*NEt_4_):* In this process, 200 mg (9.93 × 10^−5^ mol) of α-methoxy-ω-carboxylic acid PEG_42_ (M = 2015 g/mol) was dissolved in methanol (2 mL). Then, 58 μL (9.93 × 10^−5^ mol) of tetraethylammonium hydroxide solution in methanol (25 wt%) was added, and the mixture was stirred for 15 min at room temperature before the evaporation of methanol under vacuum. The crude product was solubilized in acetone (2 mL) and precipitated into diethyl ether (200 mL). The obtained white precipitate was dried under vacuum (Yield: 97%) and characterized by ^1^H NMR. ^1^H NMR (400 MHz, Acetone-*d*_6_), δ (ppm): 1.36–1.41 (m, 2H); 3.50–3.55 (d, 8H); 3.59 (s, 4x42H).

*Synthesis of tetraethylammonium* α*-maleimido-*ω*-carboxy poly(ethylene glycol) (MalPEG_62_-COO*^− +^*NEt_4_)*: In this process, 300 mg (0.1 × 10^−3^ mol) of α-maleimido-ω-carboxylic acid poly(ethylene glycol) was dissolved in methanol (2.2 mL). Then, 67 μL (0.1 × 10^−3^ mol) of tetraethylammonium hydroxide solution in methanol (25 wt%) was added, and the mixture was stirred for 15 min at room temperature before the evaporation of methanol under vacuum. The crude product was solubilized in acetone (2 mL) and precipitated into diethyl ether (200 mL). The obtained orange precipitate was dried under vacuum (Yield: 83%) and characterized by ^1^H NMR. ^1^H NMR (400 MHz, Acetone-*d*_6_), δ (ppm): 1.36–1.41 (m, 12H); 3.48–3.55 (d, 8H); 3.59 (s, 4x62H). M_NMR_ = 2860 g/mol.

*Synthesis of poly(benzyl malate) (PMLABe_73_):* PMLABe_73_ was synthesized as previously described, with slight modifications, with a monomer/initiator ratio chosen to obtain a theoretical molar mass of 15,000 g/mol (n = 73) [33]. Briefly, 16.7 mg (6.64 × 10^−5^ mol) of tetraethylammonium benzoate were solubilized into 700 μL of anhydrous ethanol freshly distilled over natrium. The initiator solution was placed in the polymerization flask, ethanol was eliminated under vacuum at room temperature and the tetraethylammonium benzoate was dried under vacuum at room temperature overnight. MLABe (1 g, 4.85 × 10^−3^ mol) of kept under N_2_ stream was transferred into the polymerization flask containing the initiator. The polymerization was conducted at 37 °C for one day (disappearance of the lactone peak at 1844 cm^−l^ from the FT-IR spectrum). The polymer was dissolved in acetone (2 mL), one drop of concentrated HCI was added and the polymer was precipitated into a large excess of ethanol (200 mL). The white precipitate was dried under vacuum (Yield: 89%) and characterized by ^1^H NMR and SEC. ^1^H NMR (400 MHz, Acetone-*d*_6_), δ (ppm): 2.97 (m, 2nH); 5.14 (m, 1nH); 5.53 (m, 2nH); 7.33 (m, 5nH); 8.04–8.06 (d, 2H). SEC (THF, 40 °C, polystyrene standards): Mw = 7,300 g/mol, Ð = 1.46.

*Synthesis of maleimide poly(benzyl malate) (MalPMLABe_73_):* In this process, 11 mg (3.33 × 10^−5^ mol) of MalHexCOO^− +^NEt_4_ was put under N_2_ stream. MLABe (0.5 g, 2.43 × 10^−3^ mol) was dissolved into 0.15 mL of anhydrous THF under N_2_ stream, and this solution was then transferred into the polymerization flask containing the initiator. The monomer/initiator ratio was chosen to obtain a theoretical molar mass of 15,000 g/mol (n = 73). The polymerization was conducted for 18 h at 37 °C (disappearance of the lactone peak at 1844 cm^−l^ from the FT-IR spectrum). The polymer was dissolved in acetone (1 mL), one drop of concentrated HCI was added and the polymer was precipitated into a large excess of ethanol (200 mL). The white precipitate was dried under vacuum (Yield: 82%) and characterized by ^1^H NMR and SEC. ^1^H NMR (400 MHz, Acetone-*d*_6_), δ (ppm): 2.97 (m, 2nH); 5.14 (m, 1nH); 5.53 (m, 2nH); 7.33 (m, 5nH). SEC (THF, 40 °C, polystyrene standards): Mw = 5730 g/mol, Ð = 1.48.

*Synthesis of poly(ethylene glycol)-block-poly(benzyl malate) (PEG_42_-b-PMLABe_73_):* In this process, 134 mg (6.65 × 10^−5^ mol) of PEG_42_COO^− +^NEt_4_ (M = 2015 g/mol) was put under N_2_ stream. MLABe (1 g, 4.85 × 10^−3^ mol) was dissolved into 0.15 mL of anhydrous THF under N_2_ stream, and this solution was then transferred into the polymerization flask containing the initiator. The monomer/initiator ratio was chosen to obtain a theoretical molar mass of 15,000 g/mol (n = 73). The polymerization was conducted for 22 h at 37 °C (disappearance of the lactone peak at 1844 cm^−l^ from the FT-IR spectrum). The polymer was dissolved in acetone (1 mL), one drop of concentrated HCI was added and the polymer was precipitated into a large excess of ethanol (200 mL). The white precipitate was dried under vacuum (Yield: 76%) and characterized by ^1^H NMR and SEC. ^1^H NMR (400 MHz, Acetone-*d*_6_), δ (ppm): 2.97 (m, 2nH); 3.58 (m, 4x42H); 5.14 (m, 1nH); 5.54 (m, 2nH); 7.33 (m, 5nH). M_NMR_ = 12,980 g/mol. SEC (THF, 40 °C, polystyrene standards): Mw = 6460 g/mol, Ð = 1.45.

*Synthesis of maleimide poly(ethylene glycol)-block-poly(β-benzyl malate) (MalPEG_62_-b-PMLABe_73_):* In this process, 104 mg (3.33 × 10^−5^ mol) MalPEG_62_-COO^− +^NEt_4_ was put under N_2_ stream, and solubilized in 0.5 mL of anhydrous THF. MLABe (0.5 g, 2.43 × 10^−3^ mol) was dissolved into 0.4 mL of anhydrous THF under N_2_ stream, and this solution was then transferred into the polymerization flask containing the initiator. The monomer/initiator ratio was chosen to obtain a theoretical molar mass of 15,000 g/mol (n = 73). The polymerization was conducted for 22 h at 27 °C (disappearance of the lactone peak at 1844 cm^−l^ from the FT-IR spectrum). The polymer was dissolved in acetone (2 mL), one drop of concentrated HCI was added and the polymer was precipitated into a large excess of ethanol (200 mL). The orange precipitate was dried under vacuum (Yield: 82%) and characterized by ^1^H NMR and SEC. ^1^H NMR (400 MHz, Acetone-*d*_6_), δ (ppm): 2.94 (m, 2nH); 3.59 (m, 4x62H); 5.14 (m, 1nH); 5.54 (m, 2nH); 7.33 (m, 5nH). M_NMR_ = 13,800 g/mol. SEC (THF, 40 °C, polystyrene standards): Mw = 8470 g/mol, Ð = 1.59.

*Grafting of peptides onto synthesized polymers MalPMLABe_73_ and MalPEG_67_-b-PMLABe_73_*: The four selected peptides (GBVA10-9, CPB, GBVA10-9scr and CPBscr) were grafted onto the maleimide end-functionalized MalPMLABe_73_ and MalPEG_62_-*b*-PMLABe_73_. Typically, (co)polymer (1 equivalent (eq.)) solubilized in 1.4 mL of DMF was added to peptide (1 eq.) solubilized in 1.4 mL of DMF. A phosphate buffer solution (PBS, pH = 7.2) was then added dropwise (54 µL of PBS at 0.01 M in 1 mL of DMF). The reaction was allowed to proceed overnight at room temperature, and excess reactant was removed by extensive dialysis against DMF for 24 h (molecular weight cut off (MWCO) of 12,000–14,000 g/mol). After DMF elimination, the purified peptide-modified (co)polymers were characterized by ^1^H NMR. The following eight peptide-modified (co)polymers were obtained: GBVA10-9PMLABe_73_, GBVA10-9PEG_62_-*b*-PMLABe_73_, CPBPMLABe_73_, CPBPEG_62_-*b*-PMLABe_73_, GBVA10-9scrPMLABe_73_, GBVA10-9scrPEG_62_-*b*-PMLABe_73_, CPBScrPMLABe_73_ and CPBScrPEG_62_-*b*-PMLABe_73_.

### 2.3. Formulation of Nanoparticles

Peptide-decorated nanoparticles (Pept-NPs) and native nanoparticles (NPs) encapsulating the fluorescence probe DiD Oil were formulated using the nanoprecipitation method initially described by Fessi et al. with slight modifications [34]. Twelve batches of Pept-NPs were thus formulated starting from 10 wt% of peptide-modified (co)polymers and 90 wt% of PMLABe_73_ or PEG_42_-*b*-PMLABe_73_ and named as indicated in Table 1.

The different batches of NPs 1, 2, 3, 7, 8 and 9 were prepared as follows: 5 µL of a solution of peptide-modified (co)polymer at a concentration of 20 mg/mL in DMF was mixed with 135 µL of a solution of (co)polymers at a concentration of 6.67 mg/mL in DMF. Then, 10 µL of a DiD Oil solution in DMF at a concentration of 0.1 mg/mL was added. This blue solution was rapidly added to 1 mL of distilled water under vigorous stirring and the resulting suspension was stirred at room temperature for 1 h. The resulting bluish suspensions were put on the top of a Sephadex G25 column. After penetration of the sample, 2.5 mL of distilled water was added. After penetration, 3.5 mL of distilled water was added and the blue NP suspensions were collected in vials and analyzed by DLS, ELS and TEM.

The NPs 4, 5, 6, 10, 11 and 12 were prepared as follows: 5 µL of a solution of peptide-modified (co)polymer at a concentration of 20 mg/mL in DMF was mixed with 135 µL of a solution of (co)polymers at a concentration of 6.67 mg/mL in DMF. Then, 10 µL of a DiD Oil solution in DMF at a concentration of 0.1 mg/mL was added. This blue solution was rapidly added to 2 mL of distilled water under vigorous stirring and the resulting suspension was stirred at room temperature for 15 min. The resulting bluish suspensions were put on the top of a Sephadex G25 column. After penetration of the sample, 0.5 mL of distilled water was added. After penetration, 3.5 mL of distilled water was added and the blue NP suspensions were collected in vials and analyzed by DLS, ELS and TEM.

In parallel, three batches of NPs were formulated starting from PMLABe_73_, PEG_42_-*b*-PMLABe_73_ and a mixture of PMLABe_73_/PEG_42_-*b*-PMLABe_73_ (Table 2).

NPs 13 were prepared as follows: 5 µL of a solution of PMLABe_73_ at a concentration of 20 mg/mL in DMF was mixed with 135 µL of a solution of PMLABe_73_ at a concentration of 6.67 mg/mL in DMF. Then, 10 µL of a DiD Oil solution in DMF at a concentration of 0.1 mg/mL was added. This blue solution was added rapidly into 1 mL of distilled water under vigorous stirring. The bluish suspension was stirred at room temperature for 1 h. NPs 14 were prepared as follows: 5 µL of a solution of PEG_42_-*b*-PMLABe_73_ at a concentration of 20 mg/mL in DMF was mixed with 135 µL of a solution of PMLABe_73_ at a concentration of 6.67 mg/mL in DMF. Then, 10 µl of a DiD Oil solution in DMF at a concentration of 0.1 mg/mL was added. This blue solution was rapidly added to 1 mL of distilled water under vigorous stirring and the resulting suspension was stirred at room temperature for 1 h. NPs 15 were prepared as follows: 5 µL of a solution of PEG_42_-*b*-PMLABe_73_ at a concentration of 20 mg/mL in DMF was mixed with 135 µL of a solution of PEG_42_-*b*-PMLABe_73_ at a concentration of 6.67 mg/mL in DMF. Then, 10 µL of a DiD Oil solution in DMF at a concentration of 0.1 mg/mL was added. This blue solution was rapidly added to 1 mL of distilled water under vigorous stirring and the resulting suspension was stirred at room temperature for 1 h. Suspensions of NPs 13, 14 and 15 thus obtained were then put on the top of a Sephadex G25 column. After penetration of the sample, 2.5 mL of distilled water was added. After penetration, 3.5 mL of distilled water was added and the blue NP suspensions were collected in vial and analyzed by DLS, ELS and TEM.

The concentration of loaded DiD oil was evaluated by UV at 660 nm using a calibration curve of DiD oil solution in a mixture of DMF/H_2_O (80/20). Then, 400 µL of the DiD oil loaded nanoparticles were mixed with 1600 µL of DMF and the resulting solution was analyzed by UV at 660 nm, giving access to the encapsulation efficiency (E.E.). The encapsulation efficiency was calculated using the following equation:E.E% = [DiD oil]_encapsulated_/[DiD oil]_initial_ × 100(1)

### 2.4. Cell Culture and In Vitro Uptake Assay of NPs

The human HepaRG hepatoma cells were grown in William’s E medium supplemented with 10% fetal calf serum (FCS), 2% L-glutamine, 5 mg/L insulin, hydrocortisone hemisuccinate at 5 × 10^−5^ M, 100 units/mL penicillin and 100 µg/mL streptomycin. The fetal calf serum (FCS) was a mixture of two different FCS batches: 75% from Biosera and 25% of FetalClone^®^ II (HyClone). The HepaRG cells were cultured at 37 °C in a humidified atmosphere of 5% CO_2_, and the medium was renewed every 2 days.

For the NPs’ uptake assay, 10^5^ HepaRG cells were seeded in 24-well plates the day before the incubations with the different batches of NPs. The culture media of the cells were renewed with media containing NPs at a final concentration of (co)polymers of 25 µg/mL. The cells were incubated with NPs for 24 h, then were detached with trypsin-EDTA and resuspended in complete medium for flow cytometry analysis and evaluation of the NPs’ uptake by measuring the fluorescence emitted by the DiD Oil-loaded NPs. For the experiments of cell uptake inhibition, cells were incubated with CPBPMLABe_73_/PMLABe_73_ (NPs 4) for 4 h in two distinct conditions: (1) at 4 and 37 °C, (2) in the absence or presence of genistein (200 μM). For the detection of fluorescent NPs within the cells by confocal microscopy, the HepaRG cells (10^4^ cells) were seeded in 8-well Lab-Tek chamber slides (cover glass slide) the day before incubation with NPs. The cells were incubated with NPs for 24 h, then, the nuclear DNA was stained with Hoechst 33342 (Sigma-Aldrich, 1 mg/mL, diluted at 1:5000) for 30 min at RT in the dark. Fluorescent NPs and nuclei were visualized by confocal microscopy.

## 3. Results and Discussion

### 3.1. Synthesis and Characterization of Peptides End-Functionalized PMLABe Derivatives

The design of nanocarriers for therapy and/or diagnosis of cancers has attracted and continues to attract huge amount of research works as highlighted by the very recent reviews published on this subject [35,36,37,38]. Such nanocarriers can take various forms (polymeric nanoparticles and micelles, inorganic nanoparticles, liposomes, etc.), be constituted by very different organic and inorganic materials, be sensitive to external (magnetic field, for example) or internal (pH variations) stimuli, and can encapsulate anti-cancer drugs and/or fluorescent probes and/or radionucleides [35,36,37,38]. However, whatever the materials used to formulate such nanocarriers, they have to respect very strict specifications: they have to be biocompatible, (bio)degradable and/or fully eliminated from the organism, to encapsulate a large amount of drugs, to specifically release their cargo at the site of action, etc. [35,36,37,38]. To fulfil such specifications, biodegradable polymeric materials must be selected to construct the nanocarriers, such as poly(amino acids) or polyesters [38]. Among the polyesters, one attracted our attention several years ago: poly(malic acid) (Figure 2) [39].

Indeed, a large family varying by the nature of lateral chains, the molar masses and/or the type of polymers (i.e., homopolymers, block/random/grafted copolymers) can be obtained by anionic ring-opening polymerization (aROP) of β-substituted β-lactones or by chemical modifications of naturally available PMLA. A wide variety of nanocarriers can therefore be formulated with properties adjusted to the targeted applications [39]. We have chosen to synthesize the selected PMLA derivatives by aROP of benzyl malolactonate (MLABe) in the presence of different carboxylate salts of tetraethylammonium as initiators (Scheme 1), because this method allows us to control both the polymers’ molar masses (fixed by the ratio monomer/initiator) and the nature of the end chains [33,39,40,41,42].

#### 3.1.1. Synthesis of PMLABe Derivatives

In the present work, four poly(benzyl malate) (PMLABe) derivatives (Scheme 1), varying by the nature of one of their end-chains (benzoate, methoxy or maleimide) and by their structure (hydrophobic homopolymers or amphiphilic block copolymers), were synthesized and characterized. While the tetraethylammonium benzoate is commercially available, the other three initiators (Figure 3) were synthesized by simply mixing the selected carboxylic acid with tetraethylammonium hydroxide.

^1^H NMR spectra of the resulting purified products allowed us to confirm that the expected initiators have been successfully synthesized and can be further used as initiators for the aROP of benzyl malolactonate (MLABe) as shown by Scheme 1. Molar mass of the PMLABe block was controlled by the ratio monomer/initiator that was adjusted to obtain PMLABe block with a theoretical molar mass of 15,000 g/mol (n = 73) [33]. After purification by precipitation in an excess of ethanol, a solvent in which both initiators and free PEG (if any) are soluble, the resulting homopolymers and block copolymers were characterized by ^1^H NMR (structure and molar mass of the PMLABe block) and size exclusion chromatography (SEC, weight average molar mass (Mw) and dispersity (Ð)). For PMLABe_73_ and MalPMLABe_73_, it was impossible to calculate its molar mass from ^1^H NMR spectra. Indeed, even if the end-chain protons were visible at 8.03–8.05 ppm (Appendix A) on the ^1^H NMR spectrum, their integration was not accurate, since these two small peaks represented only two protons over the ones of the polymer main chain. For PMLABe end-terminated by the maleimide group, the two protons of the maleimide end-groups were not, as expected, visible on the ^1^H NMR spectrum (Figure 4A).

For amphiphilic block copolymers, the molar mass of the PMLABe block was calculated using the integration of the methylene protons of the PEG block (δ = 3.59 ppm, Figure 4B and Appendix A), whose molar mass was given by the supplier, and the integration of the methylene protons of the PMLABe block (δ = 5.14 ppm, Figure 4B and Appendix A). As shown by results gathered in Table 3, calculated molar masses of PMLABe block were in good agreement with the theoretical ones fixed by the monomer/initiator ratio, thus highlighting the quite good control of the polymerization reaction [30].

Nevertheless, SEC measurements gave weight average molar masses (Mw) of the synthesized homopolymers and amphiphilic block copolymers, that are lower than the theoretical ones. We have already observed such a phenomenon for amphiphilic block copolymers [33], a phenomenon that can be attributed to the conditions used for SEC measurements (solvent and standards’ nature) together with a possible absorption of the (co)polymers onto the SEC column, leading to an underestimation of molar masses. However, SEC analyses allowed us to demonstrate the good control of the polymerization procedures because the dispersity values are lower than 1.5 (Table 3), and only one peak has been observed on all the chromatograms.

We can conclude that aROP of MLABe using the selected initiators leads to the expected (co)polymers with adequate structures, molar masses, and dispersity.

#### 3.1.2. Peptide Grafting

As widely recognized, the thiol-maleimide reaction, also known as Michael addition, allows us to easily conjugate peptides with a thiol function at either their N- or C-terminal position to polymeric materials, lipids, nanoparticles, liposomes or biomolecules, presenting an accessible maleimide group [16,43,44,45,46,47,48,49]. Such Michael addition has been described to take place under mild conditions of pH and temperature, and to be conducted in aqueous and/or organic solutions [16,43,44,45,46,47,48,49]. Moreover, maleimide functions are also well-known to have a very strong affinity and specificity towards thiol groups. Therefore, Michael addition can be realized with (macro)molecules and/or nanoobjects without reacting with other functional groups such as amines and carboxylic acids [16,43,44,45,46,47,48,49].

In this context, maleimide end-terminated PMLABe derivatives previously synthesized were further used to graft mercaptopropanoic acid C-terminus modified GBVA10-9, CPB, GBVA10-9scr (scrambled control of GBVA10-9) and CPBscr (scrambled control of CPB) peptides through Michael addition, as shown by Scheme 2.

As described in the Materials and Methods section, the Michael addition between the selected peptides C-terminated by a thiol function and the synthesized (co)polymers end-terminated by a maleimide group was conducted in DMF in the presence of phosphate buffer solution (PBS) at pH 7.2 at room temperature. The unreacted peptides were eliminated by dialysis against DMF. After elimination of DMF, peptide-grafted (co)polymers were analyzed by ^1^H NMR (Figure 5 and Appendix A).

The ^1^H NMR spectra of the purified peptide-grafted (co)polymers showed the presence of peaks corresponding to the considered peptide together with the ones corresponding to the (co)polymers. Based on the relative integration of methyl protons belonging to the peptides (0.81 ppm for GBVA10-9, 0.92 ppm for GBVA10-9scr, 0.90 ppm for CPB and 0.86 ppm for CPBscr) and of methylene groups of PMLABe at 5.14 ppm, we assumed that 80 to 95% of either PMLABe_73_ or PEG_62_-*b*-PMLABe_73_ were modified with the selected peptides (Table 4).

To make sure that free peptides were efficiently eliminated using dialysis, mixtures of PMLABe_73_ and thiol-modified GBVA10-9 and PMLABe_73_ and thiol-modified CPB, solubilized in DMF at the same concentrations as the ones used for Michael addition, were dialyzed against DMF. The solutions contained in the dialysis bags were then analyzed by ^1^H NMR after elimination of the DMF. The ^1^H NMR spectra thus obtained showed the absence of peptides’ peaks at the precision of proton NMR, while peaks corresponding to the PMLABe_73_ were observed (Appendix A).

All together, these results demonstrate the success of the peptide grafting onto maleimide end-functionalized PMLABe_73_ and PEG_62_-*b*-PMLABe_73_.

### 3.2. Preparation and Characterization of Peptides Functionalized Nanoparticles

Several methods have been described to prepare well-defined NPs starting from either hydrophobic homopolymers or amphiphilic block copolymers [50,51]. Among all the available techniques, nanoprecipitation of hydrophobic homopolymers or amphiphilic block copolymers solubilized into organic solvent miscible to water into an aqueous solution is a simple but robust method leading to well-defined NP suspensions [34,50,51]. Several years ago, we applied this technique, firstly described by Fessi et al. [34], to the formulation of hydrophobic and amphiphilic PMLABe derivatives, allowing us to reproducibly obtain well-defined NPs [33]. We therefore applied this technique to prepare peptide-decorated NPs starting from the successfully synthesized peptide-modified PMLABe derivatives. We, however, had to adjust the nanoprecipitation technique, i.e., the nature of the organic solvent and (co)polymer concentration in both the organic solvent and aqueous solution, to such PMLABe derivatives, also taking into account the fact that the fluorescence probe DiD Oil also had to be encapsulated into all NP batches for further in vitro cellular uptake studies.

As all the peptide-modified (co)polymers and DiD Oil are well-soluble in DMF (solvent miscible with water), we therefore used this organic solvent for DiD Oil-loaded peptide-decorated NP formulation. Twelve batches of peptide-decorated NP suspensions in water were thus prepared (Table 1), together with three batches of native NP suspensions (NPs without peptide, Table 2). After nanoprecipitation, the aqueous suspensions were passed through a Sephadex column to eliminate non-encapsulated DiD Oil. Indeed, Sephadex columns, usually used for proteins desalting, allow products’ separation based on steric exclusion. In this context, low-molar mass molecules remain on the top of the column, while high-molar mass compounds are eluted as a function of their molar mass: the higher the molar mass, the faster the molecule is eluted. Free low-molar mass DiD Oil is therefore retained at the top of the Sephadex column while DiD oil-loaded NPs are eluted at the exclusion volume of the column. Peptide-decorated NP suspensions were then characterized (Table 5) by dynamic light scattering (DLS) and electrophoretic light scattering (ELS), giving access to the hydrodynamic diameter (Dh), sample dispersity (PDI), and surface charge (zeta potential). Finally, DiD Oil encapsulation efficiency (E.E.) was determined by UV analysis at 660 nm (Table 5).

As shown by the results gathered in Table 5, peptide-decorated NPs have hydrodynamic diameter varying from 66 to 163 nm, while native NPs have Dh between 62 and 71 nm. Pearce et al. recently reviewed the evolution of NPs designed as cancer drug delivery systems, and more importantly they gave the adequate size range to gain satisfactory blood circulation time and adapted in vivo biodistribution. Pearce et al. thus reported that NPs should have a size between 5 and 200 nm [52]. In terms of diameter, all peptide-decorated and native NP batches therefore are in the range of nanovector’s sizes that might be used for biomedical applications [52,53]. If the majority of the NP suspensions have a moderate dispersity (PDI lower or equal to 0.25), some NP suspensions seem to be quite dispersed with PDI above 0.25 (but lower or equal to 0.30). However, the quality of these samples remains acceptable to perform in vitro assays [51]. On the other hand, all the prepared NPs showed a negative surface charge, as illustrated by negative zeta potential values varying from −37 to −59 mV (Table 5). Both the high absolute values and negative charges highlighted the high stability of the prepared peptide-decorated and native NP suspensions. Indeed, it is commonly accepted that: (i) nanovectors with an absolute value of zeta potential equal or greater than 30 mV are electrostatically stabilized, and (ii) negatively charged nanoobjects might reduce non-specific liver/spleen uptake [51,52,53]. The results gathered in Table 5 show that zeta potential values measured for NPs without peptides were slightly higher than those measured for NPs decorated with peptides. Such differences might be due to the charges present in the peptide molecules through the positively charged amino acids that are relatively abundant in all the considered peptides. These positive charges might partially neutralize the negative charges of PMLABe derivatives coming from their carboxylate end-chain. Moreover, the configuration of peptidic chains at the surface of the corresponding NPs probably plays a role on the values of zeta potentials. Indeed, peptides might be either sticking out of NPs or stuck to the surface of NPs, thus masking the negative charges coming from the (co)polymers. The hydrophobic fluorescence probe DiD Oil was efficiently encapsulated in all the prepared NP suspensions, as highlighted by the values of calculated encapsulation efficiency (E.E.) that varied from 49 to 95% (Table 5). In addition, no DiD Oil residual traces were observed on the Sephadex column for all NP suspensions, meaning that no free DiD Oil was present in the prepared suspensions.

To check the robustness of the above-described process, several batches of all peptide-decorated and native NPs were prepared and analyzed by DLS. As shown by the results gathered in Figure 6, quite reproducible results in terms of both diameters (Figure 6A) and dispersity (Figure 6B) were obtained, as shown by the reasonable error bar values.

The data given by Figure 6A also clearly show the influence of the peptide’s nature on the diameter values for NPs composed of PeptPMLABe_73_/PMLABe_73_. Indeed, a significant increase in NP diameters can be observed on one side between NPs decorated with GBVA10-9 and those decorated with its scrambled control (GBVA10-9scr), and on the other side between NPs decorated with CPB and those decorated with its scrambled control (CPBscr). Such behavior might be explained by the different physico-chemical properties of the peptides as a result of their different amino acids sequences and natures. However, more experiments need to be performed to confirm this conclusion, in particular the evaluation of the hydrophobic or hydrophilic character of each peptide. Fully PEGylated NPs decorated or not with peptides showed similar diameters with values around 80 nm, probably because of the presence of PEG chains that forced the hydrophobic inner core to be more condensed, thus lowering the influence of the peptide nature on the corresponding NPs size and/or forced the entrapment of the peptides into or close to the hydrophobic inner-core. On the other hand, quite surprising results were observed for NPs composed of the mixture Pept-PEG_62_-b-PMLABe_73_/PMLABe_73_ (10/90). Indeed, as shown by Figure 6, GBVA10-9scr, CPB and CPBscr-decorated NPs prepared from the above mentioned (co)polymers mixture have similar diameters above 120 nm, while GBVA10-9-decorated NPs based on the mixture Pept-PEG_62_-b-PMLABe_73_/PMLABe_73_ (10/90) have diameters that are close to those measured for NPs based on the mixture PEG_42_-b-PMLABe_73_/PMLABe_73_ (10/90), with values around 80 nm, similar to those observed for fully PEGylated NPs. It seems that the presence of the flexible PEG_62_ block leads to a partial entrapment of GBVA10-9 into the inner-core of the NPs because of the higher hydrophobic character of the GBVA10-9 peptide.

The morphology of both peptide-decorated and native NPs was assessed by transmission electron microscopy (TEM) (JEOL, Tokyo, Japan) using phosphotungstic acid as a staining agent (Figure 7).

TEM allows us to visualize nanoobjects’ morphology and size together with sample dispersity. As shown by TEM images, all prepared peptide-decorated and native NPs have more or less spherical shapes with dispersity similar to those measured by DLS. Diameters evaluated from TEM images were globally lower than those measured by DLS as a result of TEM samples’ preparation. Indeed, while DLS measurements considered NPs in suspensions, TEM measurements gave images of NPs under a dried state, thus leading to an under-estimation of their diameters.

### 3.3. In Vitro Internalization Assays

Peptide-functionalized NPs were next used in preliminary in vitro human hepatoma cell uptake assays. The HepaRG cells were incubated with peptide-modified NPs and native NPs at the same concentration (25 µg/mL) for 24 h, and analyzed by flow cytometry to evaluate the influence of both peptide nature and NPs composition on the cell uptake. As shown by Figure 8A, all the cells were positive regardless of the NPs’ composition, further demonstrating our previous data indicating that PMLABe-based NPs were efficiently internalized in human cells [34,40]. Nevertheless, fluorescence intensity, representing accumulation levels of NPs in HepaRG cells, significantly varied with NPs composition, the absence or presence of peptides on their surfaces, and peptides’ nature (Figure 8B). Fully PEGylated NPs (NPs 3, NPs 6; NPs 9, NPs 12, and NPs 15) were poorly internalized by HepaRG cells, even for NPs prepared with GBVA10-9PEG_62_-*b*-PMLABe_73_/PEG_42_-*b*-PMLABe_73_ and CPBPEG_62_-*b*-PMLABe_73_/PEG_42_-*b*-PMLABe_73_ copolymers. It seems that the peptides are buried into the PEG corona, as illustrated by their low average diameters measured by DLS (Table 4 and Figure 6). On the contrary, NPs constituted by CPBPMLABe_73_/PMLABe_73_ mixture (NPs 4) are highly internalized by HepaRG cells, and, as expected, the NPs with the CPB scrambled control (CPBscr, NPs 10) are significantly less internalized by HepaRG cells. Surprisingly, NPs decorated with GBVA10-9 (NPs 1) and those with GBVA10-9scr (NPs 7) on their surfaces were internalized at the same level as NPs based on PMLABe_73_ (NPs 13). GBVA10-9 and GBVA10-9scr are composed by several hydrophobic amino acids, thus probably leading to the entrapment of these peptides within the hydrophobic NPs’ inner core constituted by poly(benzyl malate).

Finally, the results obtained with NPs formulated from the mixture PeptPEG_62_-*b*-PMLABe_73_/PMLABe_73_ (10/90) are quite surprising. Indeed, both the GBVA10-9scr- and CPBscr-decorated NPs (NPs 8 and NPs 11, respectively) seem to be more efficiently internalized into HepaRG cells than similar NPs decorated with GBVA10-9 and CPB (NPs 2 and NPs 5, respectively). As already described in the literature, the nature of the linker between the peptide and the carrier (whatever its nature) has a great influence on the specificity of the peptide [54,55,56]. In this study, the linker between selected peptides and PMLABe_73_ is either the hexyl group (NPs 1, NPs 4, NPs 7 and NPs 10) or PEG_62_ (NPs 2, NPs 5, NPs 8 and NPs 11), thus probably leading to great differences in the presentation and/or presence of all the peptide amino acids sequence at the surface of concerned NPs, and therefore on the recognition ability of the considered peptide.

The cellular uptake of nanoparticles is usually an energy-dependent mechanism of endocytosis [57,58,59,60]. In order to investigate whether the uptake of peptide-functionalized NPs in HepaRG cells could also involve endocytosis (Figure 8B-insert), the energy-dependent uptake and the endocytosis were studied by comparing the uptake of CPBPMLABe_73_/PMLABe_73_-based NPs (NPs 4) at 4 and 37 °C, and in the absence or presence of the endocytosis inhibitor, genistein, in HepaRG cells. The intensity of fluorescence was strongly decreased in cells maintained in the cold at 4 °C, suggesting that the cellular entry of NPs required an energy-dependent process. In addition, the treatment with genistein also strongly reduced (~70% decrease in fluorescence) the cell accumulation of NPs 4, further demonstrating the internalization of NPs by endocytosis. Genistein is known to mainly inhibit calveolae-mediated endocytosis. Our data thus suggest that internalization of NPs 4 involves this pathway, as we previously reported for other PMLA-based NPs [61].

In addition to in vitro uptake assays, confocal microscopy studies were realized on HepaRG cells after their incubation with Pept-NPs constituted by the mixtures PeptPEG_62_-*b*-PMLABe_73_/PMLABe_73_ encapsulating the fluorescent probe DiD Oil (red fluorescence) in order to visualize the localization of Pept-NPs, the cell nucleus being stained in blue with Hoechst 33342 reagent (Figure 9). These images confirmed cell internalization of NPs with a bright red fluorescence observed as cytoplasmic dots for all NPs, with a more intense staining for NPs 11 around the cells’ nuclei.

## 4. Conclusions

Thanks to the versatility of the PMLABe synthesis method, several peptide-functionalized derivatives were successfully obtained and characterized. Two hepatotropic peptides (GBVA10-9 and CPB) and their scrambled controls (GBVA10-9scr and CPBscr, respectively) have been grafted at the end chain of two PMLABe derivatives, MalPMLABe_73_ and MalPEG_62_-*b*-PMLABe_73_, through Michael addition. Peptide-modified (co)polymers were then formulated with unmodified (co)polymers in order to introduce around 10 wt% of peptides at NPs’ surfaces. Prepared NPs showed hydrodynamic diameters, dispersity, zeta potential and morphology adapted to their uses as drug carriers in the frame of cancer therapy. Preliminary in vitro assays (flow cytometry and confocal microscopy) demonstrated the great influence of the nature of the linker between the peptide and the (co)polymer, the NPs compositions, and the nature of amino acid sequences on their uptake by HepaRG hepatoma cells. The mechanism of internalization, at least for NPs 4, involves a calveolae-dependent endocytosis. Further studies are underway to highlight the presence or absence of all the amino acid sequences at the surface of NPs: ^1^H NMR spectra in deuterated water (D_2_O) of all the formulated NPs are currently in progress. The first results obtained with NPs without peptides showed that: *(i)* for NPs based on PMLABe_73_, no peaks of polymers can be observed on the ^1^H NMR spectrum highlighting that the polymer is well-condensed as a hydrophobic core; *(ii)* for NPs based on PEG_42_-*b*-PMLABe_73_, only the peak corresponding to the PEG chains is visible on the ^1^H NMR spectrum showing the core-shell structure of such NPs; and *(iii)* for the NPs constituted by the mixture PMLABe_73_/PEG_42_-*b*-PMLABe_73_ (90/10), a small peak corresponding to the PEG chain is visible on the ^1^H NMR spectra, thus demonstrating the presence of PEG chains at the surface of these NPs, but in a smaller amount than the fully PEGylated NPs.

## Data Availability

Not applicable.

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
