# Peer review of "Synthesis of Poly(Malic Acid) Derivatives End-Functionalized with Peptides and Preparation of Biocompatible Nanoparticles to Target Hepatoma Cells"

_nanomaterials, 2021, doi:10.3390/nano11040958_

Round 1

Reviewer 1 Report

  1. Line 41, "At the beginning of the XXth century," Which century is referred in this sentence?
  2. Reference 29 is a key foundation for this work. However, it is still under review. Line 91~95 did not clearly describe how the work was done and what the work has revealed. This issue should be extensively addressed in order to support the current manuscript. 
  3. In Scheme 1, the first step involving aspartic acid is suggested to be removed. It makes the synthesis confusing - which carboxylic acid was protected by benzyl group and how the amino acid was converted to a lactone. Or the authors can add all the detailed steps. 
  4. Please add a table to summarize the conjugation percentage of these peptides on each maleimide polymer. 
  5. In Table 4, Colum Composition, please elaborate what the ratio means, i.e. weight ratio or molar ratio? Standard deviation is missing for the size of each formulation. A column for the loading efficiency is missing. 
  6. For the cellular internalization experiments, both the flow cytometry and microscope experiments are missing the DiO only control, meaning dissolving DiO stock solution in minimal DMSO percentage and evaluate the uptake. It is necessary and please do not try to address this point without adding data.
  7. What is the endocytic pathway for these peptide-conjugated particles?  

Reviewer 2 Report

The paper submitted by Brossard et al. deals with the preparation and characterization of a complete series of nanoparticles functionalized with peptides in order to target the hepatoma cells.

The manuscript is clear, well written, and the conclusions are supported by the results. However, some corrections are needed in order to increase the overall quality of the paper:

  1. The abstract section must be modified; the authors must provide the most important results and not generalities. Moreover, the term “zetametry” must be replaced with “electrophoresis” in all the manuscript, not only in the abstract.
  2. The introduction section must be completed with recent references concerning the preparation of several types of NPs functionalized with aptamers, which are an important class of ligands: https://doi.org/10.1016/j.msec.2019.109828: https://doi.org/10.3390/polym11091515; https://doi.org/10.1016/j.msec.2020.111591
  3. Lines 347-353: even if the authors stated that the SEC measurements provide smaller Mn for block copolymers this statement is generally true only for graft copolymers; for the analyzed block copolymers and especially for homopolymers, such a huge difference between Mn theo and Mn Sec cannot be attributed to the measurement conditions, maybe to some polymer adsorption on the SEC columns…However, the authors can use universal calibration method coupled to light scattering detection in order to avoid such issues.
  4. Line 374: which was the role of PBS?
  5. Lines 378-389: in order to determine the efficiency of the purification step, a NMR DOSY should be carried out. With this technique it quite easy to observe the presence of different species with different diffusion coefficients.
  6. Table 4: how the authors explain the very high negative ZP values for samples 13, 14, and 15 (without peptide) which are much higher than the peptide-functionalized samples?! A deeper discussion about the ZP values is needed at this point.
  7. Line 425: delete “e” after “and”
  8. Line 430: “quite dispersed”

Round 2

Reviewer 1 Report

The authors have properly addressed most of the comments from previous reviewers. A few more references for endocytic pathways are suggested to be included alongside Line 538 in the revised manuscript: DOI: 10.1016/j.nantod.2011.02.003; DOI: 10.1021/acs.biomac.9b01073; DOI: 10.1021/acsnano.7b02044. 

Author Response

Thank you for your valuable comments allowing to improve the quality of our manuscript. 

As suggested, we have included the three references concerning the endocytic pathways (Ref 58, 59 & 60).

Reviewer 2 Report

Can be published as it is.

Author Response

Thank you for your valuable comments allowing to improve the quality of our manuscript.